# A Bayesian Framework to Assess the Usability of Dry Powder Inhalers in a Cohort of Asthma Adolescents in Italy

**DOI:** 10.3390/children9010028

**Published:** 2021-12-31

**Authors:** Roberto Walter Dal Negro, Massimiliano Povero

**Affiliations:** 1Research & Clinical Governance, 37100 Verona, Italy; robertodalnegro@gmail.com; 2AdRes Health Economics and Outcome Research, 10121 Torino, Italy

**Keywords:** dry powder inhalers, bronchial asthma, usability, global usability score, adolescents

## Abstract

The useability of DPIs (dry powder inhalers) depends on several factors that are influenced by the patients’ subjectivity and objectivity. The short-form global usability score (S-GUS), a specific tool for the quick ranking and comparison in real life of an inhaler’s usability, was used to investigate six of the most prescribed DPIs (Breezhaler, Diskus, Ellipta, Nexthaler, Spiromax, and Turbohaler) in consecutive asthma patients aged <18 years. A Bayesian indirect comparison (IC) was carried out to merge all pairwise comparisons between the six DPIs. Thirty-three subjects participated: eighteen tested Breezhaler, Spiromax, Nexthaler, and Ellipta simultaneously, while fifteen tested Breezhaler, Spiromax, Diskus, and Turbohaler. The estimates of the S-GUS, by the IC model, allowed us to rank the DPIs by their degree of usability: Ellipta, Diskus, and Spiromax were classified as “good to pretty good” (S-GUS > 15), while Spiromax, Turbohaler, and Breezhaler were classified as “insufficient” (S-GUS < 15). The multidomain assessment is recommended in asthma adolescents in order to approximate the effective usability of different DPIs as best as possible. The S-GUS proves particularly suitable in current clinical practice because of the short time required for its use in adolescents.

## 1. Introduction

The adherence to inhalation therapy, together with the personalization of respiratory treatments, have been strongly supported in the last two decades for persistent bronchial asthma needing the long-term use of active drugs, and particularly in subjects prone to insufficient compliance to inhalation, such as the elderly and adolescents [1,2,3]. 

Dry powder inhalers (DPIs) became progressively available as a result of technological engineering [4,5,6]. However, patients are unable to use all inhalers equally well, independently of their age [2,6]. Consequently, inhaler use still represents a critical issue because it may affect therapeutic outcomes, regardless of the molecules prescribed [4,5,6,7,8,9]. 

The factors affecting patient adherence to inhalation treatments via DPIs have been extensively studied. In particular, they have been mostly investigated from the point of view of patients, in terms of their preference, acceptance, and satisfaction [10,11,12,13,14,15], while specific studies aimed at assessing the correspondence between patient beliefs, DPI performances, and their effective usability in real life have been episodic [16,17,18].

The global usability score (GUS) is an anonymous questionnaire specifically developed for objectively assessing the usability of inhalation devices [19]. The GUS questionnaire was selected, with respect to other instruments available in the literature [15,16,17,18], because it allows for the assessment of a wider range of usability indicators, including the usability cost. While data concerning the usability of different DPIs measured by means of the GUS questionnaire are already available for adult asthmatics, corresponding data have not yet been provided for adolescents.

The study aimed to assess and compare the real-life usability of six of the most used DPIs in asthma adolescents.

## 2. Materials and Methods

The short-form GUS questionnaire (S-GUS) was preferred for the present study because, while maintaining the same sensitivity of the extensive version, it is much faster to fill (mean time 6.1 vs. 23.4 min; *p* < 0.001) [20], and it was thus considered much more suitable for adolescents. The S-GUS was administered to a sample of consecutive subjects, aged <18 years, and who had been treated with long-term inhalation therapy for persistent asthma at the CEMS Specialist Centre (Verona-Italy) over the period from October–December 2018. According to a previous analysis [20], the mean S-GUS of DPIs associated with a good usability is 26 (95% CrI 21 to 32), the mean S-GUS of DPIs associated with a pretty good usability is 20 (95% CrI 15 to 25), and for DPIs judged insufficient, the mean S-GUS is 11 (95% CrI 8 to 13). For each pair-wise comparison (good vs. pretty good, good vs. insufficient, and pretty good vs. insufficient), we calculated the expected sample size assuming a 5% type I error, 80% statistical power, and a clinically significant difference equal to the value that resulted in that comparison. The final sample size was chosen as the maximum value between the sample sizes calculated for each comparison. According to this algorithm, at least 28 patients should be enrolled in the study.

The S-GUS questionnaire, reported in Appendix A, consists of twelve questions and it quickly allows for the objective assessment and comparison of the usability of, at the most, four different devices simultaneously [20]. Patients had to answer the first nine questions before DPI utilization, and the last three questions after DPI utilization. The answers to the first nine items were scored by a single subscore each. Items #10 and #11 were scored by decreasing values, according to the degrees of difficulty objectively encountered by the patients with each DPI, while a categorical subscore was assigned to Item #12, such as 3.2 (Yes), or 0 (No), in case of agreement or disagreement between the patient’s and the nurse’s judgment, respectively (see Appendix A). The maximum value of the S-GUS is 50 points, and higher values are associated with greater usability.

The six DPIs to compare were: Breezhaler, Diskus, Ellipta, Nexthaler, Spiromax, and Turbohaler. These DPIs were selected because they are characterized by different intrinsic resistances (ranging from 0.017–0.039 kPa^0.5^ L/min) [21,22], and also because they require a variable number of actions for their actuations (7 for Breezhaler, 4 for Turbohaler, and 3 for the remaining devices). 

The methodology was previously described [20]. Briefly, two nurses, trained for more than two years, supervised all of the patient procedures for inhalation. For each session, the nurse assessed, firstly, the previous experience with devices (Items #1–3), and they explained how to use each DPI in random order. Patients were asked to declare their preference “at a glance” according to six prespecified DPI characteristics (Items #4–9). Finally, the patient had to prepare the actuation autonomously (for each device), and the nurse monitored and assessed the execution (Items #10–12). 

The patient characteristics were reported as absolute and relative frequencies; age was the only continuous variable collected in the questionnaire, and it was summarized as the mean and standard deviation (SD). The nonparametric Wilcoxon test was used to check the differences in the age distribution between the subgroups of patients who tested the different sequences of DPIs, while the Fisher’s exact test was used for all the other variables. The analyses were conducted using the statistical software R version 3.6.3 (R Core Team, Vienna, Austria), and a *p*-value lower than 0.05 was considered to indicate evidence of differences in the evaluated variables.

A Bayesian indirect comparison (IC) was conducted to merge all pairwise comparisons between the six DPIs reported in the 33 questionnaires. The model was developed by using the software package, WinBUGS 1.4.3 (MRC Biostatistics Unit, University of Cambridge, UK) [23], and was based on the hierarchical random effect model, described in Dias et al. [24]: $$\{\begin{array}{c} Y_{\mathrm{ij},s}\sim N\left(\delta_{\mathrm{ij}}{+u}_{s}{,\;\sigma }_{\mathrm{ij},s}^{2}\right)\\ u_{s}\sim N\left({0,\tau }^{2}\right){,\;s=1,\ldots ,N}_{\mathrm{studies}} \end{array},$$
where $Y_{\mathrm{ij},s}$ in the S-GUS of Device i compared with Device j, according to the s-th patient; $\sigma_{\mathrm{ij},s}^{2}$ is the variability of the Y_ij,s_ for Patient s; $\delta_{\mathrm{ij}}$ is the average effect of Device i compared with Device j; and u_s_ are random terms that captures the heterogeneity among patients (quantified by the variance, τ^2^). The relative device usability, $\delta_{\mathrm{ij}}$, is expressed in terms of the absolute mean difference (AMD). 

Noninformative vague priors for all the hyperparameters defined in the model were used: normal distribution centered in zero with a large variance (10^4^) for the device usability and uniform distributions, with supports between 0 and 5 for all the standard deviations. All the simulations were conducted using one chain of 100,000 iterations, and the estimates were calculated after a 50,000-run burn-it. The parameters of interest were summarized by the posterior mean and the 90% credibility intervals (CrI), i.e., the 5th and 95th percentiles of the simulated values. 

The Bayesian approach allows for establishing an efficacy ranking among the analyzed interventions. The surface under the cumulative ranking (SUCRA) [25] was calculated, and the devices were ranked from the best (highest SUCRA) to the worst (lowest SUCRA). Finally, the sources of variability were evaluated by the posterior distribution of the proportion of variability due to the between-patient heterogeneity (I^2^), according to the classical definition of Higgins and Thompson [26].

## 3. Results

A total of thirty-three consecutive asthma adolescents were included in the study: eighteen patients (Group 1) tested Breezhaler, Spiromax, Ellipta, and Nexthaler, and fifteen patients (Group 2) tested Breezhaler, Spiromax, Turbohaler, and Diskus. 

The general characteristics of the sample are reported in Table 1. About half of the subjects were already familiar with DPIs, while one third had previous experience with metered dose inhalers (MDIs), and 6% had experience with soft mist inhalers (SMIs) (Table 1). 

Group 1 and Group 2 were similar for almost all collected characteristics (Table 1). The proportion of males was slightly lower in Group 1 (39% vs. 67%, *p* = 0.107). 

Figure 1 describes the network obtained from the 33 questionnaires. Eighteen patients in Group 1 (represented with medium thick lines) tested only Ellipta, Nexthaler, and Breezhaler, and fifteen patients in Group 2 (represented with thin lines) tested only Turbohaler and Diskus. All the patients in both groups tested both Breezhaler and Spiromax (represented with a thick line).

The GUS values estimated for each device are summarized in Table 2, ranked from the most (highest S-GUS) to the least usable (lowest S-GUS) DPI. Ellipta was the most usable device (mean S-GUS = 25.5), with a SUCRA = 99.4% (Figure 2). Diskus almost tied Ellipta (mean S-GUS = 25.1), with a SUCRA = 80.6%, and Spiromax was ranked third (mean S-GUS = 20.6), with a SUCRA = 60%. Among the least usable devices, Turbohaler was ranked fourth (mean S-GUS = 15.3), with a SUCRA = 37.9%, and Nexthaler fifth (mean S-GUS = 12.2), with a SUCRA = 22%. Finally, Breezhaler was ranked last (mean S-GUS = 9.0), with a SUCRA = 0.1%.

According to the thresholds previously proposed [20], Ellipta, Diskus, and Spiromax proved to have “good to pretty good” usability (S-GUS > 15 points), while Turbohaler, Nexthaler, and Breezhaler showed “insufficient” usability (GUS < 15 points) (Figure 3).

Finally, the S-GUS score values proved large indeed, with a total heterogeneity (i.e., total variance) equal to 24.81. However, the patients’ characteristics seem to explain almost 70% of the variability (I^2^ = 67.4%). 

The main result was that the higher GUS scores were clearly in favor of DPIs that proved easier to actuate, regardless of their volume, shape, appearance, color, and attractivity to adolescents, and independently of patient subjectivity.

## 4. Discussion

The prescription of the most suitable DPIs to use is still a critical issue [1,2], particularly in adolescents because they are usually less compliant than adults to regular inhalation treatments. 

In recent years, the patient’s point of view has increasingly come to be valued in terms of preference, and the majority of studies mainly focus the role of criteria strictly on the opinions of patients. The willingness to use, the “at a glance” preference, satisfaction, intuitivity, and acceptance were the most adopted criteria for comparing different DPIs [10,11,12,13,14,15,16,27].

Moreover, factors independent of patient opinions can further contribute to the assessment of the so-called “usability” of inhalation devices, and of DPIs in particular. For instance, it has been proven that different regimens of the intrinsic resistances of DPIs are associated with different performances in terms of the lung deposition of the respiratory drugs and, consequently, of the clinical outcomes, even if patients are totally unaware of their relative role [5,13,21,22,28].

This evidence is supported by 90% of health professionals in the United Kingdom when affirming that they are really concerned about the possible occurrence of problems if the use of the inhalation device is not specified in the prescription, and that they are absolutely convinced that DPIs are noninterchangeable in the same patient, unless required by motivated reasons [28,29]. 

Usability is the result of a combination of subjective (i.e., intuitivity, satisfaction, willingness to use, “at a glance” preference, acceptance) and objective (the device’s characteristics, independent of patient convictions and beliefs and the costs included) domains. It should be considered as the discriminating parameter for choosing the proper DPI, as it defines the overall “real-word” convenience of each DPI. This was the main reason for adopting the multidimensional GUS questionnaire in the present study on asthma adolescents. 

Even if all the DPIs tested in the present study did not achieve the GUS top score of 50 points, their usability proved clearly ranked in asthma adolescents, simplifying their choice in real life. Moreover, the patients’ usability elaborated in the present study is consistent with those observed in a cohort of asthmatic adults previously instructed in the use of DPIs [30]. The ranking observed in the adult cohort was qualitative comparable to the ranking observed in the present cohort of asthma adolescents. This could suggest that the age of the patient is not a factor that is independently associated with the usability of DPIs. 

The present study has some limitations. The study is a monocentric investigation with a small sample size. Each patient did not test all the selected DPIs simultaneously, but only four-by-four. On the other hand, this was decided on purpose because the simultaneous comparison of more than four DPIs would have affected the reliability of the patient responses. Actually, despite the strict control of the nurses’ explanations of each DPI procedure, it was possible that minimal differences occurred while delivering the educational messages. 

## 5. Conclusions

DPI usability is confirmed as a complex issue indeed. DPIs are characterized by different degrees of usability in asthma, and also in adolescents. One single parameter only, based on the patient’s perceptions and beliefs, is unable to exhaustively represent per se the usability of DPIs, and allow for the most effective choice. A multidimensional score should be preferred in asthma adolescents in order to approximate the effective usability of the different DPIs presently available at the best possible level. The current conviction that “one size fits all” is clearly misleading in clinical practice when using DPIs, particularly in asthma adolescents. The opportunity to quickly rank the usability of DPIs should be regarded as a value, and should then be carefully considered before prescribing or switching a DPI in asthma adolescents.

## Figures and Tables

**Figure 1 children-09-00028-f001:**
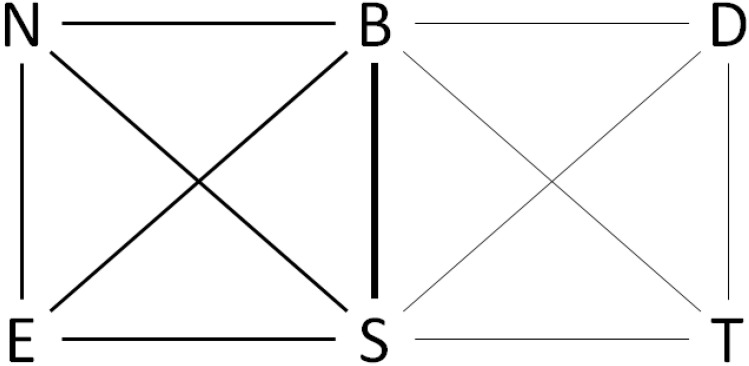
Evidence network based on patients in the two groups; medium thick lines represent pairwise comparisons available in Group 1 (*N* = 18), thin lines represent pairwise comparisons available in Group 2 (*N* = 15), and thick line represents pairwise comparison available in both groups (*N* = 33). B: Breezhaler; D: Diskus; E: Ellipta; N: Nexthaler; S: Spiromax; T: Turbohaler.

**Figure 2 children-09-00028-f002:**
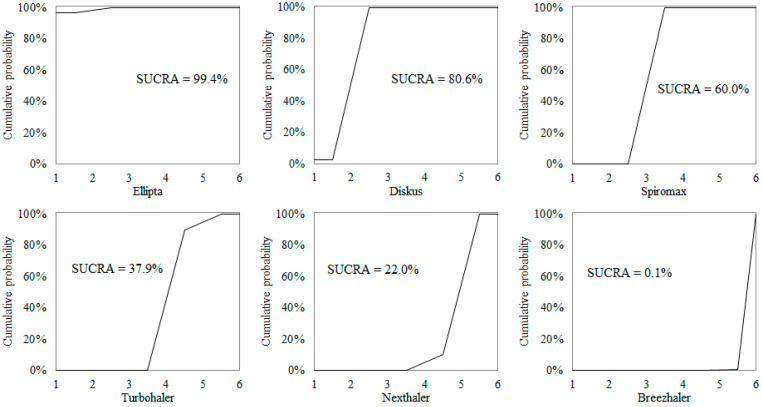
Cumulative ranking probability plots from the six DPIs.

**Figure 3 children-09-00028-f003:**
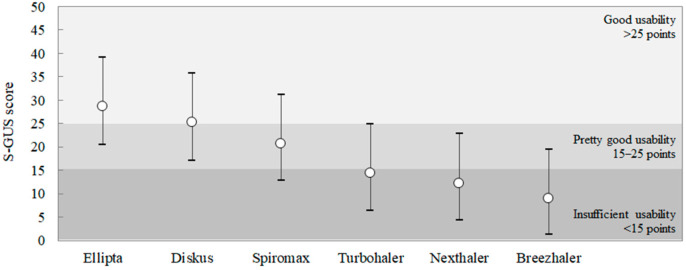
The thresholds for ranking DPI usability by means of the estimated S-GUS scores: white points represent the mean S-GUS, while the bars indicate the 90% credibility interval.

**Table 1 children-09-00028-t001:** Baseline characteristics of patients and evaluated devices using the S-GUS questionnaire.

	Total (*N* = 33)	Group 1 (*N* = 18)	Group 2 (*N* = 15)	*p*-Value
Instructed to DPIs (%)	17 (51.5%)	8 (44.4%)	9 (60.0%)	0.295
Instructed to MDIs (%)	11 (33.3%)	5 (27.8%)	6 (40.0%)	0.355
Instructed to SMIs (%)	2 (6.1%)	1 (5.6%)	1 (6.7%)	0.71
Mean age (SD)	16.3 (2.47)	16.4 (2.05)	16.4 (1.69)	0.621
Male (%)	17 (51.5%)	7 (38.9%)	10 (66.7%)	0.107
Country (%)				0.199
North	25 (75.8%)	15 (83.3%)	10 (66.7%)	
Center	3 (9.1%)	0 (0.0%)	3 (20.0%)	
South and Islands	5 (15.2%)	3 (16.7%)	2 (13.3%)	
Education (%)				0.391
Lower secondary	26 (78.8%)	15 (83.3%)	11 (73.3%)	
Upper secondary	7 (21.2%)	3 (16.7%)	4 (26.7%)	

DPIs: dry powder inhalers; MDIs: metered dose inhalers; SD: standard deviation; SMIs: soft mist inhalers.

**Table 2 children-09-00028-t002:** Results of the Bayesian analysis on the six DPIs analyzed.

**Ellipta S-GUS = 28.5** **90% CrI (20.6 to 39.2)**	Post *p* = 0.970	Post *p* = 1.000	Post *p* = 1.000	Post *p* = 1.000	Post *p* = 1.000
AMD = 3.44 90% CrI (0.4 to 6.5)	**Diskus S-GUS = 25.1** **90% CrI (17.2 to 35.8)**	Post *p* = 0.999	Post *p* = 1.000	Post *p* = 1.000	Post *p* = 1.000
AMD = 7.92 90% CrI (5.6 to 10.2)	AMD = 4.48 90% CrI (2.1 to 6.8)	**Spiromax S-GUS = 20.6** **90% CrI (12.9 to 31.2)**	Post *p* = 1.000	Post *p* = 1.000	Post *p* = 1.000
AMD = 14.26 90% CrI (11.4 to 17.1)	AMD = 10.82 90% CrI (8.3 to 13.3)	AMD = 6.34 90% CrI (4.2 to 8.5)	**Turbohaler S-GUS = 14.3** **90% CrI (6.4 to 25.0)**	Post *p* = 0.897	Post *p* = 1.000
AMD = 16.32 90% CrI (14.0 to 18.6)	AMD = 12.88 90% CrI (10.0 to 15.7)	AMD = 8.40 90% CrI (6.4 to 10.4)	AMD = 2.06 90% CrI (−0.6 to 4.7)	**Nexthaler S-GUS = 12.2** **90% CrI (4.4 to 22.9)**	Post *p* = 0.997
AMD = 19.52 90% CrI (17.3 to 21.7)	AMD = 16.08 90% CrI (13.8 to 18.4)	AMD = 11.60 90% CrI (10.0 to 13.2)	AMD = 5.26 90% CrI (3.1 to 7.4)	AMD = 3.20 90% CrI (1.3 to 5.1)	**Breezhaler S-GUS = 9.0** **90% CrI (1.4 to 19.5)**

DPIs on the diagonal are reported in order according to S-GUS ranking. Comparisons between S-GUS DPIs should be read in the lower triangular area from left to right, and the estimate is in the cell in common between the column-defining DPI and the row-defining DPI (AMDs below 0 favor the column-defining DPI). Posterior probabilities (Post *p*) should be read in the upper triangular area from left to right, and the estimate is in the cell in common between the row-defining DPI and the column-defining DPI (high posterior probability indicates high credibility of results). AMD: Absolute mean difference; CrI: credibility interval; S-GUS: short-form global usability score.

## Data Availability

The authors do not wish to share the data without their permission.

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
