# Peer review of "A Bayesian Framework to Assess the Usability of Dry Powder Inhalers in a Cohort of Asthma Adolescents in Italy"

_children, 2021, doi:10.3390/children9010028_

Round 1

Reviewer 1 Report

The paper is overall well written. However, the key problem I noticed here is a number of participants, that is relatively low (33 for 6 different devices, despite a good model of testing in methodology), so maybe paper could be a kind of "pilot study". Prevalence of asthma in childhood and adolescence is very high, so I think there is no problem to have more participants in the study (at least more than 100).

Author Response

Reviewer

The paper is overall well written. However, the key problem I noticed here is a number of participants, that is relatively low (33 for 6 different devices, despite a good model of testing in methodology), so maybe paper could be a kind of "pilot study". Prevalence of asthma in childhood and adolescence is very high, so I think there is no problem to have more participants in the study (at least more than 100).

ANSWER. First, thank you very much indeed for defining the paper “well written”. DPIs Usability of patients suffering persistent respiratory disease was already previously investigated in pretty large cohorts: 148 adult subjects with asthma (Assessing the Global Usability of Dry Powder Inhalers: Analysis of Six Devices Widely Used for Asthma, DOI:10.24966/PMRR-0177/100064) and 103 adult subjects with COPD (Patients’ usability of seven most used dry-powder inhalers in COPD, https://doi.org/10.1186/s40248-019-0192-5). In the present study we wanted to investigate if adolescents and adults (checked in our previous study) were somehow different in terms of GUS response. Despite the small sample, results seem to be quite credible because they proved very similar to those observed in adults. Moreover, the Bayesian approach was chosen as the most suitable and recommended for analysis of small samples  (Innovative research methods for studying treatments for rare diseases: methodological review, DOI: 10.1136/bmj.g6802). However, we added the small sample size in the limitations.

Reviewer 2 Report

This paper is very well conducted. Few questions to the authors: 1)During the study the tested devices were original samples (with active drugs) or with placebo. Considering that the use of different drugs with LABA was there any concern regarding side effects? Were the patients aware of it? 2) Was there any differences related to previous experience with the different devices?

Author Response

This paper is very well conducted. Few questions to the authors: 

1) During the study the tested devices were original samples (with active drugs) or with placebo. Considering that the use of different drugs with LABA was there any concern regarding side effects? Were the patients aware of it? 

ANSWER.

First, thank you for defining the paper “well conducted”. Devices were completely empty of any active drug. On the other hand, the study was aimed to investigate the usability of different DPIs independently of their content.

2) Was there any differences related to previous experience with the different devices?

ANSWER. In our previous paper focused on adult asthmatic patients we observe that DPIs’ usability was different between patients with previous experience with DPI and naïve patients (“Assessing the Global Usability of Dry Powder Inhalers: Analysis of Six Devices Widely Used for Asthma”, DOI: 10.24966/PMRR-0177/100064). In particular, the usability of Breezhaler seems not associated with the patient’s original level of previous instruction as it was characterized by the lowest GUS score both in the experienced and in the naïve group. Furthermore, in the DPI experienced group, the trend in usability proved quite linear from Ellipta (with the highest GUS) to Breezhaler (the lowest GUS), whereas, in the naïve group, GUS scores of almost all devices (with the exclusion of Turbohaler ranked first) resulted very similar and also 95% CrI were very close. However, due to the small sample, we decided to not include this variable in the present analysis and to postpone the investigation on the possible association between previous DPIs’ experience and GUS in  future larger studies.
